# Pollution Weather Prediction System: Smart Outdoor Pollution Monitoring and Prediction for Healthy Breathing and Living

**DOI:** 10.3390/s20185448

**Published:** 2020-09-22

**Authors:** Sharnil Pandya, Hemant Ghayvat, Anirban Sur, Muhammad Awais, Ketan Kotecha, Santosh Saxena, Nandita Jassal, Gayatri Pingale

**Affiliations:** 1Symbiosis Centre for Applied Artificial Intelligence, Symbiosis International (Deemed) University, Pune 412115, Maharashtra, India; head@scaai.siu.edu.in; 2Innovation Department, Technology University of Denmark, Copenhagen 2800, Denmark; hemgha@dtu.dk; 3Symbiosis Institute of Technology, Symbiosis International (Deemed) University, Pune 412115, Maharashtra, India; anirban.sur@sitpune.edu.in (A.S.); saxena.santosh.btech2018@sitpune.edu.in (S.S.); nandita.jassal.btech2018@sitpune.edu.in (N.J.); gayatri.pingale.btech2018@sitpune.edu.in (G.P.); 4Centre for Intelligent Medical Electronics, Department of Electronic Engineering, School of Information Science and Technology, Fudan University, Shanghai 200433, China; muhammada17@fudan.edu.cn

**Keywords:** healthy living, variability analysis, pollution prediction, linear regression, artificial neural network, smart outdoor monitoring

## Abstract

Air pollution has been a looming issue of the 21st century that has also significantly impacted the surrounding environment and societal health. Recently, previous studies have conducted extensive research on air pollution and air quality monitoring. Despite this, the fields of air pollution and air quality monitoring remain plagued with unsolved problems. In this study, the Pollution Weather Prediction System (PWP) is proposed to perform air pollution prediction for outdoor sites for various pollution parameters. In the presented research work, we introduced a PWP system configured with pollution-sensing units, such as SDS021, MQ07-CO, NO2-B43F, and Aeroqual Ozone (O_3_). These sensing units were utilized to collect and measure various pollutant levels, such as PM2.5, PM10, CO, NO_2_, and O_3_, for 90 days at Symbiosis International University, Pune, Maharashtra, India. The data collection was carried out between the duration of December 2019 to February 2020 during the winter. The investigation results validate the success of the presented PWP system. In the conducted experiments, linear regression and artificial neural network (ANN)-based AQI (air quality index) predictions were performed. Furthermore, the presented study also found that the customized linear regression methodology outperformed other machine-learning methods, such as linear, ridge, Lasso, Bayes, Huber, Lars, Lasso-lars, stochastic gradient descent (SGD), and ElasticNet regression methodologies, and the customized ANN regression methodology used in the conducted experiments. The overall AQI values of the air pollutants were calculated based on the summation of the AQI values of all the presented air pollutants. In the end, the web and mobile interfaces were developed to display air pollution prediction values of a variety of air pollutants.

## 1. Introduction and Related Works

The high-paced economy, moving onwards because of construction, agricultural activities, industrial plants, and many other factors, enables development and poses risks to the human race. In such scenarios, it is essential to maintain a healthy life quality that includes some necessary activities, such as having a balanced diet and keeping the body moving. With that, it is also evident to keep the body hydrated to let out harmful toxins continuously. A lot of diseases can be prevented by just changing your style of living, as mentioned above. The Global Burden of Disease (GBD) study addressed some risk factors, such as smoking, high blood pressure, unsafe sex, alcohol use, air pollution, etc. In 2017, this report mentioned that 34.1 million deaths and 1.21 million DALYs (disability-adjusted life years) were attributed to GBD risk factors [1]. Many factors can affect human health and reduce life expectancy. However, still, one cannot neglect that air pollution is one of the leading causes of various severe diseases, like lung cancer and heart diseases, and accounts for One-third of these deaths. Household air pollution is counted as one of the worst, as it has particulate matter and causes stroke, ischemic heart disease, chronic obstructive pulmonary disease (COPD), and many more. According to the state of the Global Report (2019), air pollution is the fifth-leading risk factor for mortality worldwide. It also reduces the life expectancy (on average) by one year and eight months globally. The report suggests that it is necessary to have a deep understanding of the key sources that provide air pollution to move forward and air pollution control policies [2]. World Health Organization reports (WHO Reports, 2020) state that air pollutants can be microscopic and reach the human system with ease. Air pollutants can vary in size. Some of these particles can be extremely fine. Hence, the particles with small diameters (PM2.5 or smaller) and ultrafine particles with a diameter of one micron or less can easily penetrate tissues and organs. Another critical topic is premature deaths, the major leading cause being household air pollution through smoke from cooking fires, burning dung, wood, coal, etc. According to World Health Organization reports (WHO Reports, 2019), the countries with low or middle incomes breathe the highest amounts of polluted air, and more than 80% of urban populations breathe air that exceeds the WHO guidelines of air quality [3,4]. P. Mamta and Bassin J. K. conducted a study in the year 2009 that showed that the overall pollution index of three different areas in Delhi City falls under “poor” and “very poor” due to respirable suspended particles, respectively [5]. Kyrkilis et al. discussed an approach to measure the aggregated air quality index (AAQI) for 18 different cities. However, the presented approach did not discuss anything about air pollution prediction [6]. A. Kumar and P. Goyal examined the AQI (air quality index) and estimated it by examining the concentration of air pollutants [7]. Zhang et al. analyzed the fine particular matter (PM2.5) characteristics at the city level in China [8]. Reche et al. discussed an air quality monitoring approach for European cities. However, they did not discuss any ideas related to air pollution prediction [9]. Anenberg et al. analyzed the characteristics of pollutants, such as black carbon and methane. However, they did not discuss any ideas related to air pollution monitoring or prediction [10]. Xu et al. discussed an ensemble learning model to predict PM2.5. However, the presented approach did not predict pollutants, such as PM10, CO, NO_2_, and O_3_ [11]. Vicente et al. discussed a new quality index for a variety of pollutants. However, they did not provide any details for air quality monitoring and prediction [12]. Sowlat et al. examined that many research types are being carried out on the AQI because air pollution is increasing at an infinite pace [13]. P. Chen discussed real-time monitoring of the urban environment. However, he did not discuss any concepts related to air pollution prediction [14]. Gurjar et al. brought forward several ways to determine the air quality index, the first being the traditional method, which includes finding the air quality by using instruments at fixed monitoring stations scattered all over the city [15].

Jiang et al. discussed various air quality monitoring approaches using social media. However, they did not research the area of air pollution prediction [16]. Meng et al. analyzed various air pollution-related risks and their impacts in Shanghai, China [17]. Xi et al. discussed results to enhance the AQI precision [18]. Petroleum-Gas University conferred and analyzed various methodologies, such as artificial neural networks, genetic algorithms, decision trees, k-nearest neighbor, and logistic regression, which have been successfully used in air quality prediction problems [19]. Suo et al. examined and researched several scenarios concerning variable AQIs to analyze the current environmental energy system’s impacts. The results indicated that PM2.5 is a pollutant that should be handled to maintain air quality for a better and safer living situation. These researches can help the authorities adjust and build strong energy structures, mitigation strategies, etc. [20]. Shen et al. introduced that the commonly used air quality index (AQI) does not entirely display pollutants’ health effects [21]. Al-Ali et al. and Devrakonda et al. designed a GPRS-controlled monitoring system, but it lacks air pollution monitoring and prediction aspects [22,23]. Marjovi et al. discussed mobile sensor network-based approaches for urban environments. The proposed methods also made use of remote sensing and spatial variability analyses. However, the proposed plans did not include air pollution prediction and monitoring [24]. Abraham et al. discussed an approach to measure indoor pollutants such as CO, O_3_, and CO_2_. However, the proposed method was not tested at outdoor sites [25]. Antonic et al. analyzed a wearable sensing-based system to measure air quality via cloud computing technology [26]. U. Kumar and V. Jain discussed an approach to forecasting ambient air pollutants using an auto regression-based exponential analysis. However, the proposed method was not tested at outdoor sites for accuracy and reliability [27]. Al-Haija et al. examined a microcontroller-based gas monitoring approach. However, the proposed method was focused on general-purpose gas sensing [28]. Shi et al. discussed a system that can do a spatial variability analysis of PM2.5 via network stations. However, the presented method was not applied to other air pollutants, such as CO, PM10, temperature, humidity, and NO_2_ [29]. According to an Atlan report, a company called “SocialCops” designed a system to measure Delhi’s air pollution. The system used a variety of air pollutant sensors but did not facilitate air pollution prediction [30]. Egondi et al. analyzed the timely variation of PM2.5 concentration levels of Nairobi and Kenya. However, the proposed approach was not applied to a variety of air pollutant parameters [31]. Shi et al. conducted an extensive survey of Northern China regions’ pollution processes using a reanalysis meteorology methodology. The proposed approach also made use of remote sensing and spatial variability analyses. However, the proposed system did not include air pollution prediction and monitoring [32]. Zhang et al. discussed the PM2.5 concentration characteristics in five diverse regions of China [33]. Marques et al. developed a system using wired sensing technology. The proposed method was not focused on any air quality monitoring or prediction aspects [34]. Kim et al. calculated the benefits of reducing PM10 and its effects on humans’ healthy living around South Korea [35]. In recent times, fellow researchers [36,37] attempted to use a moving vehicle to measure air pollutions’ variations using mobile sensor networks. However, the proposed approach was not tested with multiple air pollutants and did not discuss air pollution prediction.

Hog-di et al. discussed an approach to monitor dust particles present in the surrounding atmosphere and its basic predictions. However, the proposed method could not facilitate direct measurements of air pollutants [38]. Richards et al. developed a wired sensing network to provide real-time updates of air pollutants using GUSTO (Generic Ultraviolet Sensors Technologies and Observations) sensing. However, the proposed approach did not offer wireless connectivity and lacked technologies such as the Internet of Things (IoT) and wireless sensing [39]. Reisinger et al. utilized optical absorption methodology for measuring a variety of air pollutants. However, the undertaken study did not facilitate air pollution monitoring and prediction for real-time data [40]. Tiwari et al. presented a microcontroller-based system to monitor methane variations. The proposed method did not facilitate air quality monitoring or prediction [41]. Dhingra et al. put up a basic design to monitor air quality parameters and a mobile application using cloud computing technology. However, the presented system was not tested at outdoor sites and did not perform accuracy and reliability-related assessments [42]. Li et al. discussed an approach that could do constant monitoring of Taiwan using 71 fixed air pollution monitoring stations. However, the conducted research did not present any air quality prediction framework for air pollution monitoring and prediction [43]. Kumar et al., Ferdoush et al., Bacco et al., and Morawska et al. analyzed air pollution monitoring approaches using widely known Zigbee technology [44,45,46,47]. However, the presented system lacked air pollution prediction and was not suitable for long distances, and its implementation was very costly. Sharma et al. proposed an MSP430 controller-based approach to measure ammonia using carbon nanotube (CNT)-based sensors. The presented approach did not discuss any transmission, monitoring, or prediction approach [48]. Barot et al. examined a QoS (Quality of Service) that enabled an air pollution framework for monitoring air pollutants. However, the proposed approach did not facilitate air pollutants’ air pollution prediction using artificial intelligence (A.I.)-based technologies. This work was also coauthored by Pandya S., who represents the presented research work [49]. Pandya et al. discussed a smart sanitizer tunnel to disinfect humans. However, it did not discuss air pollution monitoring and prediction [50]. Kularatna et al. analyzed an Arduino-based system to measure LPG (Liquefied Petroleum Gas). However, the proposed approach did not emphasize air quality monitoring and prediction aspects [51]. According to NAMP reports, India’s government proposed a program called “National Air Quality Monitoring,” which provided data about various pollutants via the www.aqi.in the portal via 779 networking stations. However, the NAMP (National Air Quality Monitoring Programme) did not offer air pollution prediction and failed to reach its developed or urban parts [52,53]. Therefore, it is essential to consider the issue of pollution prediction based on real-time data. 

Recently, previous studies have conducted extensive research in the areas of air pollution and air quality monitoring. Despite this, the fields of air pollution monitoring and prediction remain plagued with unsolved issues concerning incineration, metallurgical process, miscellaneous, and many more. In this study, a pollution weather prediction (PWP) system is presented that can perform air pollution predictions for outdoor sites using various pollution parameters for healthy breathing and living. In the undertaken study, we collected real-time data of air pollutants such as PM2.5, PM10, CO, NO_2_, and O_3_ using a variety of sensing units, such as the SDS021 sensing unit, the MQ07-CO sensing unit, the NO2-B43F sensing unit, and an Aeroqual Ozone (O_3_) sensing unit. The data was collected from an outdoor site situated at Symbiosis International University, Pune, Maharashtra, India. The data collection was carried out between the duration of December 2019 to February 2020 during the winter. The detailed description of the data collection process is described in Section 2.2.3. The investigation results validate the success of the presented PWP system. In the conducted experiments, linear regression and artificial neural network (ANN)-based AQI predictions were performed for 60,000 air pollution samples. In the end, the overall AQI values of air pollutants were calculated based on the summation of AQI values of all the presented air pollutants. The proposed approach can function lifelong and can be retrained using updated pollution data in the future.

The proposed paper is organized as follows: Section 2 presents the pollution weather prediction system’s necessity and discusses the detailed PWPS workflow. Section 3 depicts a detailed analysis and discussion of the investigated results of the PWS system and represents a web and mobile interface representation of the presented PWP system. The concluding remarks and the possibility of future enhancements are discussed under Section 4. The Appendix A describes the essential concepts such as air quality monitoring, air quality index (AQI), air monitoring stations, and a list of terminologies used in the experiments.

## 2. The Necessity of the PWP System

The presented PWP system’s significant contributions are as follows: (i) An IoT-based framework is proposed to perform real-time data collection of various air pollutants. (ii) Detailed variability analysis of air pollutants such as PM2.5, PM10, CO, NO2, and O3 is conducted in the presented research work. (iii) An AI-based machine learning and an in-depth neural network-based approach are proposed to perform the air pollutant assessment and prediction of AQI values of air pollutants. In the conducted experiments, linear regression and ANN-based AQI prediction were performed. Furthermore, the presented study also found that the customized linear regression methodology outperformed other machine-learning methods, such as the linear [54], ridge [55], Lasso [56], Bayes [57], Huber [58], Lars [59], Lasso-lars [60], stochastic gradient descent (SGD) [61], and ElasticNet [62] regression methodologies, and the customized ANN regression methodology used in the conducted experiments. (iv) In the end, the web and mobile interface was developed to display the air pollution prediction values of a variety of air pollutants. The investigation results validated the success of the presented PWP system.

### 2.1. Details of the Controller and Sensors

#### 2.1.1. ESP8266 12E/NodeMCU Controller

ESP8266 12E/NodeMCU is a cost-effective controller module equipped with a 32-bit microcontroller with 4MB memory. According to [58], a NodeMCU controller consists of GPIOs, I2C, UART, ADC, and PWM pins to interface with various sensor modules. Figure 1a represents a microcontroller unit used in the conducted experiments. Table 1 represents a comparative analysis of various controllers available in the market. 

#### 2.1.2. SDS021 Particulate Matter Sensing Unit

The SDS021 sensing unit can detect and measure 2.5-µm and 10-µm dust particles from the surrounding environment. It is based on the UART protocol and provides a bit rate of 9600 bits/second. The measurement range of the SDS021 sensing unit is 0–100 mg/m^3^. The SDSO21 sensing unit module contains a fan for keeping a consistent wind stream over the detection chamber. The presented sensing unit required a 5-V power backup and a 60-mA current. The SDS021 sensing unit required a 4-mA current in sleep mode. Table 2 shows the technical specification details of the SDS021 sensing unit. Figure 1b represents the SDSO21 sensing unit used to measure the PM2.5 and PM10 concentration levels.

#### 2.1.3. MQ07-CO Carbon Monoxide (CO) Sensing Unit

The MQ07-CO sensing unit is capable of measuring CO levels with the 9600-bit rate. It is also based on the UART protocol, which facilitates a range between 0–1000 ppm (parts per million). Figure 1c represents the MQ07-CO sensing unit used to measure the carbon monoxide concentration level. Table 3 illustrates the technical specification details of the MQ07-CO sensing unit.

#### 2.1.4. NO_2_-B43F Nitrogen Dioxide Sensing Unit

NO2-B43F is a low-cost sensing unit that can sense nitrogen-oxide from the surrounding environment. It facilitates strong signal levels along with zero current capability. It allows a resolution up to 50 parts per billion (ppb) and an operating range of 0.001 to 0.02 ppm. Figure 1d represents the NO2-B43F sensing unit, which is used to measure the NO_2_ concentration levels. Table 4 describes the technical specification details of a NO2-B43F sensing unit.

#### 2.1.5. Aeroqual Ozone (O_3_) Sensing Unit

Aeroqual Ozone O_3_ is a low-cost sensing unit that was designed to provide consistent and accurate ozone levels. It has a measurement range of 0.002–0.01 ppm. Figure 1e represents an O_3_ sensing unit used to measure the ozone (O_3_) concentration levels for 1-h and 4-h durations. Table 5 illustrates the technical specification details of an Aeroqual Ozone (O_3_) sensing unit.

### 2.2. Layered Architecture of an IoT-Based Air Quality Monitoring System

The layered architecture of the presented PWP system was classified into seven sub-components: (i) physical sensing layer, (ii) communicating and sensing layer, (iii) cloud service layer, (iv) data preprocessing layer, (v) data visualization layer, (vi) data prediction layer, and (vii) application layer. Figure 2 represents the layered design of a PWP system.

#### 2.2.1. Physical Sensing Layer

The physical layer consists of the sensors, which measure the pollutants present in the atmosphere. The pollutants measured are Particulate Matter 2.5 (PM2.5), Particulate matter 10 (PM10), ozone (O_3_), nitrogen dioxide (NO_2_), and carbon monoxide (CO). Figure 3 represents the system architecture of a PWP system. The detailed working of a PWP system is described in the remaining sections.

#### 2.2.2. Communication and Networking Layer

The communication layer establishes a link between the physical sensing layer and the backend. It provides the interface due to which the data can be extracted. Figure 4 represents a detailed communication workflow of a communication layer. As described in the physical layer, pollution sensors connected with a NodeMCU controller measure the pollutants present in the surrounding environment. As shown in Figure 3, a PWP system uses the MQTT protocol (MQTT 5.0) and a HIVEMQ cloud broker to establish an interface between a publisher and a subscriber [63]. In the conducted experiments, the “paho-MQTT” and “paho.mqtt.Client” packages were used (MQTT 5.0, accessed on 20 November 2019). MySQL is responsible for storing the data acquired by the data acquisition module. Finally, the PWP ASP.Net-based web application and a mobile application fetch the pollutant data of various pollutants such as PM2.5, PM10, CO, NO_2_, and O_3_ from the MySQL repository and represent in the form of pollution reports.

#### 2.2.3. Data Analysis Layer

The data analysis layer is responsible for handling the data collection, data preprocessing, and data visualization. The dataset used in the conducted experiments is mainly divided into three categories: (i) Pollutants: the PWP system dataset contains various pollutants such as CO, NO_2_, PM2.5, PM10, and O_3_, concerning their AQI values. (ii) Locations: GPS location coordinates such as latitude and longitude. (iii) Timestamp: timestamp parameters such as date and time. The researchers and governments use an air quality index (AQI) metric to identify the current pollution conditions at a particular location. The air quality index (AQI) is a useful color-coded unitless index that effectively communicates air pollution concentrations to the general public [64,65]. The AQI metric is already used in certain countries such as Malaysia, Canada, and Europe as a primary pollution classification metric. The computation of the AQI of a particular pollutant can be done based on metrics such as the average period obtained from the PM sensing units. The classification risk metric is employed in the conducted experiments based on the AQI values of various pollutants such as PM2.5, PM10, CO, NO_2_, and O_3_. As shown in Table 6, the classification of the risk metrics is described in various categories: good, moderate, unhealthy, very unhealthy, and hazardous. The AQI coefficient can be given by [64],
(1)$$\text{AQI}\;\text{coefficient}\;\text{of}\;\text{a}\;\text{particular}\;\text{pollutant}\;=\;\left[\frac{(P_{obs}-P_{min)\;}(AQI_{max}-AQI_{min)\;}\;}{P_{max}-P_{min}}\right]$$
where $P_{obs}$ = average measured concentration in 24 h in mg/m^3^, $P_{max}=$ the maximum concentration of an AQI of a particular pollutant calculated based on the risk classification metric, $P_{min}=$ the minimum concentration of an AQI of a particular pollutant calculated based on the risk classification metric, $AQI_{min}$ = minimum AQI values of a specific pollutant calculated based on the risk classification metric. The molecular conversion coefficient (CMC) to convert a pollutant from ppb to µg/m^3^ can be given by [64]
(2)$$\mathrm{CMC}\left(\mu g/m^{3}\;\right)=\frac{\left[\left(\mathrm{ppb}\right)\times \mathrm{mw}\right]}{\mathrm{mv}}$$
where mw = molecular weight and mv = molecular volume. The molecular volume (mv) can be given by [64]
(3)$$\mathrm{mv}\left(\mathrm{in}\;\mathrm{litres}\right)=\frac{\left[22.41\;\times \;T\;\times 1013\right]}{\left(273\;\times p\;\right)}$$
where T = temperature (K) and P = atmospheric pressure (hPa (hectopascal)).

##### Data Preprocessing

As shown in Figure 3, in the presented research work, we introduced a PWP system that is configured with pollution sensing units such as SDS021, MQ07-CO, NO2-B43F, and Aeroqual Ozone (O_3_). These sensing units were utilized to collect and measure various pollutant levels, such as PM2.5, PM10, CO, NO_2_, and O_3_. The data was collected from an outdoor site situated at Symbiosis International University, Pune, Maharashtra, India. The data collection was carried out between the duration of December 2019 to February 2020 during the winter. After collecting the pollutant data, an AQI was calculated based on the acquired pollutant data, as shown in equation (1) for various pollutants such as PM2.5, PM10, CO, NO_2_, and O_3_. In the experiments, the quantities of pollutants are independent, and the AQI of a particular pollutant is a dependent variable. In the undertaken study, the PWP dataset consists of 189,648 training samples and 18 features. The missing values are majorly present in the pollutants section. If we eliminate a training sample with at least one missing value, then a 50% data loss is expected. Due to many instances, we kept the training samples with missing values for further use. The proposed PWP system was trained with 60 k examples divided into training and testing phases. The rest of the data was used for validation testing. The feature selection dimensionality reduction data preprocessing technique was applied to the PWP dataset to select the presented system’s required features. In the PWP dataset, the dependent features are the AQI of the pollutants such as PM2.5, PM10, NO_2_, CO, and O_3_, whereas the independent features are the quantities of PM2.5, PM10, NO_2_, CO, and O_3_. Therefore, it is tough to apply feature extraction here; hence, the feature selection technique was used on the PWP dataset to reduce the data dimensions. In total, 12 features were selected (six dependent variables and six independent variables). During data preprocessing, the standard-scaler scaling technique was applied in the presented research work.

a. Data Visualization

In the undertaken study, a histogram-based analysis was conducted to check the variability of various pollutants. The variability of the pollutants was set between 0 to 100 μm, approximately. Figure 5a,b represents the linear relationship representation of PM2.5, PM10, CO, NO_2_, O_3_ (1 h), and O_3_(4 h) concerning their AQI values. It can be observed that the AQI values of the pollutants are directly proportional to the number of air pollutants. The AQI of a particular pollutant (α) can be given by
(4)α ∝ the quantity of a particular air pollutant 
(5)AQI of a particular pollutant= σ × quantity of air pollutants+c
where *σ* = X-intercept/slope and c = constant; c can be a nonzero value if the *x*-axis starts with a nonzero term. Here, equation (4) and (5) will formulate a hypothetical line of air pollutants such as PM2.5, PM10, CO, NO_2_, O_3_ (1 h), and O_3_ (4 h). Since time is a categorical value and fed to the PWP system, it may be considered a relative value that can affect the PWP system’s overall accuracy.

Figure 6a,b represents the variability analysis of various air pollutants such as PM2.5, PM10, CO, NO_2_, O_3_ (1 h), and O_3_ (4 h). Here, the *x*-axis represents time, and the *y*-axis represents the contamination values of the air pollutants. The air pollutants’ variability analysis is described using various graphical representations such as line graphs, scatter graphs, and histograms. The visual representation of line graphs and scatter charts indicates the linear relation of the pollutants concerning time, whereas the histogram depicts the pollutants’ variability. Figure 6a shows slight variability when the scale is reduced to 0–25 μm. Furthermore, PM10 ranges from 2 to 12-+ μm. PM2.5 ranges from 2 to 8 μm. The rest of the pollutants like CO, NO_2_, O_3_ (1 h), and O_3_ (4 h) have values from 0 to 1 μm. Figure 6b represents the variability of different gases such as PM2.5, PM10, and CO. It indicates that the variability of CO exceeds the frequency of 140 marks; it was also recorded that CO-based air contamination remained zero for the maximum time of the day, with less variability recorded throughout the day. Furthermore, it could be observed that PM2.5-based air contamination remained around 35 μm, with high variability recorded throughout the day. PM10-based air contamination remained about 105 μm for the day’s maximum time, with high variability recorded throughout the day. Figure 6b also represents a variety of air pollutants’ values concerning time. It can be observed that the PM2.5, PM10, and CO average contamination values were recorded around 0 to 35 μm, 0–7 μm, and 0–115 μm, respectively.

#### 2.2.4. Data Prediction Layer

The data prediction layer is responsible for predicting AQI values of various air pollutants using linear regression methodology. The PWP dataset labeled data such as the AQI values and quantities of a particular pollutant. The samples used in the conducted experiments were continuous and predictable. Table 7 represents the details of the data collection for the conducted experiments.

(a) Linear Regression based prediction: 

In the first phase of the presented system, linear regression methodology was applied to achieve a hypothetical line of air pollutants, as shown in Figure 9. The predicted AQI coefficient ($Y_{predicted}$) of a single pollutant can be given by
(6)$$\text{Predicted}\;\text{AQI}\;\text{Coefficient}\;(Y_{predicted})\;=\;\left(\sigma \;\times x\right)+c$$
where σ= slope parameter, x = quantity of a particular pollutant, and c = predicted AQI coefficient constant; c can be a nonzero value if the *x*-axis starts with a nonzero term.
Predicted AQI = y-predicted quantity if a particular pollutant

Now, to calculate the error between the $Y_{predicted}$ and $Y_{actual}$ values, the cost function j (σ ) can be given by
(7)$$j\;\left(\sigma \;\right)\;=\;\frac{\left({\;Y}_{predicted}-{\;Y}_{actual}\right)}{2\;\times m}$$
where m = the number of training examples. To achieve the value of the minimum cost function j (σ ) and to calculate the slope parameter σ, a gradient descent methodology-based optimization function can be represented by a gradient descendant graph of j (σ ), as shown in Figure 7.

To obtain the optimum global value of a cost function j (σ ), we applied the repeated convergence of a gradient descent technique as below:$$\begin{array}{l} Repeat\;until\;convergence\\ \{\\ \;temp\;+=\;\sigma \;-\;\alpha \;\frac{\partial \left(j\;\left(\sigma \;\right)\right)}{\partial \sigma }\\ \;\sigma =temp\\ \} \end{array}$$
where α= learning rate and temp = a temporary variable. If a slope parameter σ is ahead of a derivative ($\frac{\partial \left(j\;\left(\sigma \;\right)\right)}{\partial \sigma }$) value, then the derivative of a negative slope will be –ve; otherwise, the value of a slope parameter σ will be gradually incremented and shift towards the optimum global value. If an optimum global value is behind, then the derivative ($\frac{\partial \left(j\;\left(\sigma \;\right)\right)}{\partial \sigma }$) the value will be +ve. Therefore, the value of a slope parameter σ will gradually decrease and shift towards the optimum global value. The applied learning rate α will be in the range of 0.001. If a learning rate α is high, then it may skip the optimum global value. If a learning rate α is low, it will require additional time to reach the optimum global value. After applying the gradient descent methodology, a slope parameter σ can be calculated, and the predicted AQI coefficient ($Y_{predicted}$) of a particular air pollutant can be given by
(8)$$\begin{array}{c} \text{Predicted}\;\text{AQI}\;\text{Coefficient}\;\left(Y_{predicted}\right)\;=AQI\;of\;PM2.5+\;AQI\;of\;PM10+\\ \;AQI\;of\;CO+\;AQI\;of\;NO2+\;AQI\;of\;O3 \end{array}$$
(9)$$\begin{array}{c} \text{Predicted}\;\text{AQI}\;\text{Coefficient}\;\left(Y_{predicted}\right)\;=\left(\sigma \;*\;\text{NO}2\right)\;+\;\left(\lambda \;*\text{CO}\right)\;+\;\left(\;\mu \;*\;\text{PM}2.5\right)\;+\\ \;\left(\phi \;*\;\text{PM}10\right)\;+\;\left(\xi \;*\;O3\right)\;\ldots \; \end{array}$$

Similarly, all the remaining pollutants were calculated. After the completion of the training process, the PWP prediction system provided the following parameter values:
*σ* = 734.33, *λ* = 11.08, *μ* = 4.00, *ϕ* = 2.00, *ξ* = 943.78,

Therefore, the final predicted AQI coefficient ($Y_{predicted}$) can be given by
(10)$$\begin{array}{c} \text{Predicted}\;\text{AQI}\;\text{Coefficient}\;\left(Y_{predicted}\right)\;=\left(734.33\;*\;\text{NO}2\right)\;+\;\left(11.08\;*\;\text{CO}\right)\;+\;\\ \left(4.00\;*\;\text{PM}2.5\right)\;+\;\left(2.00\;*\;\text{PM}10\right)\;+\;\left(943.78\;*\;O3\right)\;+\;0 \end{array}$$

Similarly, in general, the predicted AQI coefficient ($Y_{predicted}$) can also be represented by
(11)$$\text{Predicted}\;\text{AQI}\;\text{Coefficient}\;\left(Y_{predicted}\right)\;=\sum_{n=1}^{6}W\;\left(n\right)\;\times x\left(n\right)$$
where W = weight values of the parameters of air pollutants, *n* = number of air pollutants, x=quantities of pollutants of N, and N represents a set of air pollutants = {PM 2.5, PM 10, CO, NO2, O3} (12).

(b) Artificial Neural Network-based Prediction: 

In the second phase of the presented system, artificial neural network methodology is applied to predict various air pollutants’ hypothetical line. The mathematical model was used to build the new equation to estimate the PWP system for air pollutants such as NO_2_, PM2.5, PM10, CO, and O_3_. An ANN is a machine-learning model that follows some of the observed genetic nervous classification properties and draws on adaptive genetic erudition. In short, to solve a problem using ANN, the following steps must be taken:
Select the type of machine-learning network for the kind of regression problem to be resolved, such as identifying a PWP system for an air pollutant gas estimator. One of the best solutions for that purpose is to use a multilayered machine-learning perceptron network.Data preprocessing: The data gathered for this study consists of a set of 189,648 training sample instances; each instance being composed of six variables. All variables were initially expressed in ppm, such as NO_2_, PM2.5, PM10, CO, O_2_, and O_3_. The sample data was later normalized into [0.1, 0.9] the interval, which streamlined the ANN model’s learning.ANN Model design: There are numbers of machine-learning architectures to select from, each having its particular parameters and compensations for its specific problem. One of the most widespread machine-learning models is the feed-forward multilayered perceptron. In this network, the ANN model consists of 7 layers: an input layer, five hidden layers, and an output layer. The hidden layers contain 512, 512, 256, 128, and 64 neurons, respectively. The final layer includes a single output neuron. The hidden layers consist of all the probable networks between the input to the output layer and allow for a combined impact of multiple independent variables on the output layer. The first input layer relates to the independent variables (NO_2_, PM2.5, PM10, CO, O_2_, and O_3_), while the final output layer corresponds to the dependent variable score of the PWP system for air pollutant gases. To achieve the final architecture mixture, such as setting the size and number of the hidden layers, we were accompanied by a cross-validation method. The architecture used in the undertaken study is a customized multilayered ANN model, as shown in Figure 8. To build a customized ANN model to predict the AQI values of various pollutants, various parameters such as the activation function, error functions, and optimizers must be configured. In the undertaken study, the rectilinear unit (ReLU) was used as an activation function; a mean absolute and mean squared error was chosen as the error function. Furthermore, adam was chosen as an optimizer. In the end, the number of neurons and hidden layers were selected based on the size and features of the dataset, as represented in Table 7. To set the weights for all the nodes in the ANN, a backpropagation technique was employed. The learning rate was set to 0.2, and the momentum term was 0.3. The ANN training was stopped when the mean squared errors (MSE) reached below 0.001.


## 3. Results and Discussion

In the undertaken study, we presented two approaches to predict a variety of air pollutants, such as PM2.5, PM10, CO, NO_2_, and O_3_: (i) the linear regression-based approach (ii) ANN-based approach.

### 3.1. Air Pollution Prediction Using a Linear Regression Methodology

In a PWP linear regression-based air pollution prediction methodology, the gradient descent is used as an optimization algorithm, as discussed in Section 2.2.4. Figure 9a–f represents a predicted AQI value of the PM2.5, PM10, CO, NO_2_, O_3_ (1 h), and O_3_ (4 h) air pollutants. The *X*-axis represents the air pollutants’ quantities, and the *Y*-axis represents the air pollutants’ AQI values. The presented results consist of three features: (i) a hypothetical line, (ii) raw data, and (iii) test data. The hypothetical line is generated by the presented customized linear regression-based air pollution prediction system. The raw data are represented with red color; the test data are the data used for the predictions depicted by a cross-blue shade structure. The overall AQI is generated after the summation of all the AQI values of the air pollutants. The parameters of NO_2_, CO, O_3_ (1 h), O_3_ (4 h), PM2.5, and PM10 are 734.33, 11.08, 943.78, 1176.2, 4, and 2, respectively. The lowest value of the x-intercept is two, which is of PM10 pollutants. It indicates that a high increment in PM10 may cause only a slight change in the AQI. The PM2.5 x-intercept is 4, which means that a high increment in a pollutant may cause less increment in an AQI value than PM10. The x-intercept of CO is 11.08, which means that a high increment in pollutants may cause less increment in an AQI but more as compared to PM2.5 and PM10. The x-intercept of NO_2_ is 734, which means that a low increment of NO_2_ may cause a high increment of an AQI value. The x-intercepts of O_3_ (1 h) and O_3_ (4 h) have the maximum readings, i.e., 943.78 and 1176.20, which means a slight increment in O_3_ cause a considerable increment in an AQI as compared to NO_2_. Each value depicts a different conclusion. PM10 has the lowest value of x-intercept, i.e., 2, which indicates that even an enormous increment in PM10 cannot cause a significant change in the overall AQI. The x-intercept of PM2.5 pollutants is 4, meaning that a high increment in this pollutant value can cause a little increment in the AQI value compared to PM10. The x-intercept of CO is 11.08, which indicates that a high increment in the pollutant value may cause a slight increment in the overall AQI; in comparison to PM2.5 and PM10, this increment caused by CO is considered more significant. The x-intercept of NO_2_ is 734, indicating that even a low increment in NO_2_ may cause a high increment in the overall AQI value. The x-intercepts of O_3_ (1 h) and O_3_ (4 h) have the maximum readings, i.e., 943.78 and 1176.20, which means that even a slight increment in O_3_ may cause a considerable increment in the overall AQI as compared to NO_2_ and others.
(12)X-intercept [increment] ∝ LR-AQI [increment]
where order (causing minor changes to significant changes) = O_3_, NO_2_, CO, PM2.5, and PM10. Table 8 represents the accuracy metrics of a linear regression methodology for a variety of air pollutants. It indicates that the maximum accuracy is achieved in the case of PM2.5 and PM10; the recorded accuracy is around 99.01 and 98.12, respectively. The CO pollutant recorded the third-highest accuracy, which is approximately 98%. Similarly, the NO_2_ pollutant recorded 91% accuracy. Lastly, the O_3_ pollutant recorded an almost similar accuracy for O_3_ (1 h) and O_3_ (4 h), which were around 96.21% and 96.59%. Based on the conducted experiments, we also found that the customized linear regression methodology outperformed the other machine-learning methodologies, such as the linear [1], ridge [2], Lasso [3], Bayes [4], Huber [5], Lars [6], Lasso-Lars [7], SGD [8], and ElasticNet [9] regression methodologies, as shown in Table 10.

### 3.2. Air Pollution Prediction Using Artificial Neural Networks

In a PWP ANN-based air pollution prediction methodology, ReLU was used as an activation function, and adam was used as an optimization function with an RMSE loss function, as described in Section 2.2.4. Figure 9a–f also represents an ANN predicted AQI value of PM2.5, PM10, CO, NO_2_, O_3_ (1 h), and O_3_ (4 h) air pollutants. The *X*-axis represents quantities of air pollutants, and the *Y*-axis represents the AQI values of air pollutants. Color coding was used to describe three features: (i) hypothetical line, (ii) raw data, and (iii) test data. As shown in Figure 9a, a gradual incline was noticed in the AQI value of a PM2.5 pollutant based on the conducted experiments. Similarly, in the case of a PM10 pollutant, a minor incline was observed compared to the PM2.5 pollutants. Figure 9c depicts a concentration level of CO pollutants and indicates a lesser scale as compared to the PM2.5 and PM10 pollutants. Furthermore, in the case of a CO pollutant, a greater inclination was recorded than the particular matter pollutants. As shown in Figure 9d, the dispersion rate of the NO_2_ pollutants was very high compared to the other pollutants. In the case of NO_2_, a huge inclination was noticed according to the graphical representation scale. As shown in Figure 9e,f, the highest inclination was recorded in the O_3_ pollutant (1 h) and O_3_ pollutant (4-h) graphs.
(13)X-intercept [increment] ∝ ANN-AQI [increment]
where order (causing minor changes to significant changes) = O_3_, NO_2_, CO, PM2.5, and PM10.

A deep neural network generates a hypothetical line. Furthermore, the raw data is represented with a red color, and the test data is represented using a cross-shade blue color structure. In the end, the overall AQI values were calculated based on the summation of the AQI values of all the presented air pollutants. Table 9 represents the accuracy metrics of an ANN algorithm for PM2.5, PM10, CO, NO_2_, O_3_ (1 h), and O_3_ (4 h). It indicates that the PM2.5 pollutant recorded the maximum accuracy as compared to all the other pollutants. Furthermore, for the rest of the pollutants, such as the PM10, CO, NO2, O_3_ (1 h), and O_3_ (4 h) pollutants, a recorded accuracy below the rudimentary level was recorded less than 85%. It also concludes that the customized linear regression methodology outperformed the ANN algorithm’s customized version in the conducted experiments. However, the NO_2_ and CO pollutants recorded around 84% and 83% accuracy, respectively. In the cases of PM2.5 and PM10, the documented accuracy was around 90% and 84%, respectively. Nevertheless, pollutants such as O_3_ (1 h) and CO recorded the lowest accuracy, which was approximately 79% and 77%.

Table 10 represents the accuracy metrics comparison of the linear [1], ridge [2], Lasso [3], Bayes [4], Huber [5], Lars [6], Lasso-Lars [7], SGD [8], ElasticNet [9], and ANN regression methodologies. It also indicates that linear regression methodology recorded the maximum accuracy, which was around 99% among all the other methods. Furthermore, it can also be observed that the customized version of the ANN regression methodology used in the conducted experiments recorded the minimum accuracy for a variety of air pollutants, as compared to the remaining machine-learning methodologies. It recorded 90%, 84%, 83%, 84.79%, 79% and, 77% accuracy for the air pollutants such as PM2.5, PM10, CO, NO_2_, O_3_ (1 h), and O_3_ (4 h), respectively. Furthermore, Table 11 represents the error metrics comparison of the linear [1], ridge [2], Lasso [3], Bayes [4], Huber [5], Lars [6], Lasso-Lars [7], SGD [8], ElasticNet [9], and ANN regression methodologies. It can be observed from the obtained error calculations that the linear regression methodology recorded the very minimal mean absolute and mean squared errors, which were around 1.12% and 3.22%, respectively. Furthermore, it also indicates that the customized ANN methodology used in the conducted experiments recorded the minimum accuracy for a variety of air pollutants, such as PM2.5, PM10, CO, NO_2_, O_3_ (1 h), and O_3_ (4 h), as compared to the remaining machine-learning methodologies. It recorded the maximum mean absolute and squared errors around 11.23% and 21.63%, respectively.

### 3.3. PWPS Web and Mobile Interface

We designed a web and mobile interface for the presented pollution weather prediction system based on the conducted experiments. Figure 10a,b represents a web and mobile interface of a PWP system. The proposed study was carried out to provide the pollution predictions of various air pollutants, such as PM2.5, PM10, CO, NO_2_, and O_3_. The application layer of the conducted study consists of a web application called “PollutionWeatherPredictionSystem.” This application presents the data collected, such as the concentration levels of various air pollutants. The application works pretty simply yet efficiently; the user interacts with the website through a calendar control that takes a specific date as the input, fetches the corresponding data from the database on MySQL, and displays it to the user, as shown below. Figure 10c represents a date-wise pollution prediction of 13 June 2020 to 24 June 2020.

## 4. Conclusions and Future Work Discussion

In the presented research work, we proposed an economic pollution weather system to predict various air pollutants for outdoor sites. The proposed PWP system consists of five layers. The physical sensing layer consists of different sensing units of air pollutants connected to a connection layer via a HIVEMQ (MQTT-based) broker. The data analysis layer performs a variability analysis of various air pollutants and discusses the variability analysis results. In the presented research work, we used machine learning and deep neural network-based approaches to predict the AQI values of various air pollutants, such as PM2.5, PM10, CO, NO_2_, and O_3_. In the future, the presented method can function lifelong and can be retrained according to the updated dataset. Based on the conducted experiments, we also found that the customized linear regression methodology outperformed other machine-learning methods, such as the linear, ridge, Lasso, Bayes, Huber, Lars, Lasso-Lars, SGD, and ElasticNet regression methodologies, and the customized ANN regression methodology used in the conducted experiments. In the case of linear regression methodology, it was observed that, whenever there were slight increases in the pollutants such as PM2.5, PM10, and CO, the overall AQI was not affected too much. However, the overall AQI value was significantly impacted whenever there was some increase in the NO_2_ and O_3_ pollutants. The overall AQI value was calculated based on the summation of all the presented air pollutants’ AQI values. In the end, the web and mobile interface was developed to display the air pollution prediction values of a variety of air pollutants. In the future, fellow researchers can research places with high altitudes, geographical variations, and spatiotemporal variations.

## Figures and Tables

**Figure 1 sensors-20-05448-f001:**
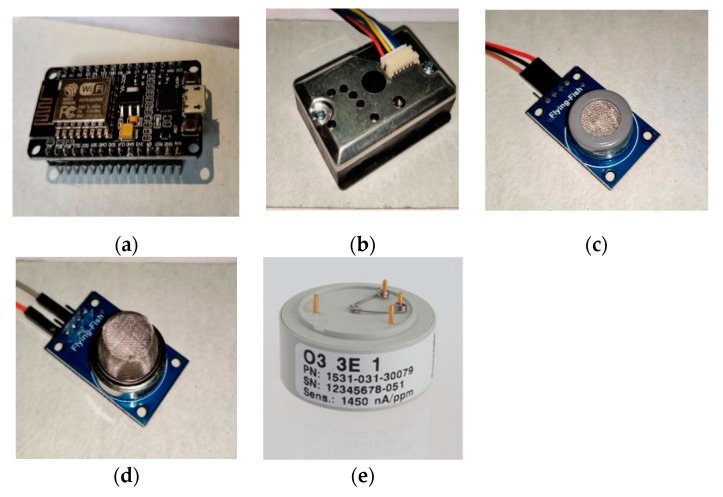
Pollution weather prediction (PWP) system sensing unit arrangements. (**a**) NodeMCU microcontroller unit. (**b**) SDSO21 (PM2.5 + PM10) sensing unit. (**c**) M.Q. 07 monitoring carbon monoxide sensing unit. (**d**) NO_2_ sensing unit. (**e**) O_3_ sensing unit.

**Figure 2 sensors-20-05448-f002:**
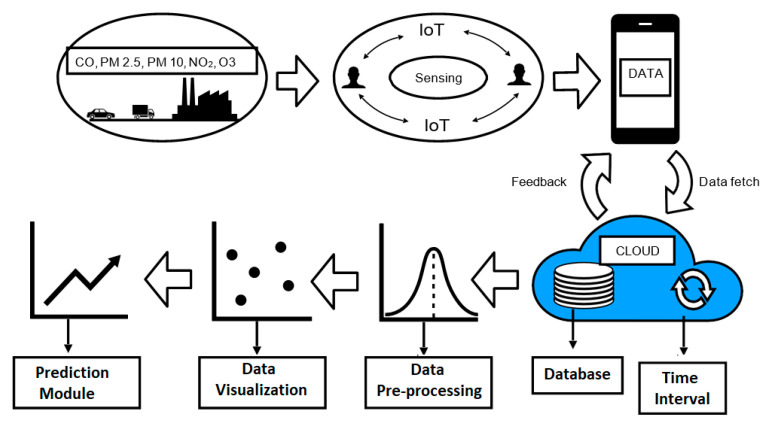
The layered architecture of the PWP system. IoT: Internet of Things.

**Figure 3 sensors-20-05448-f003:**
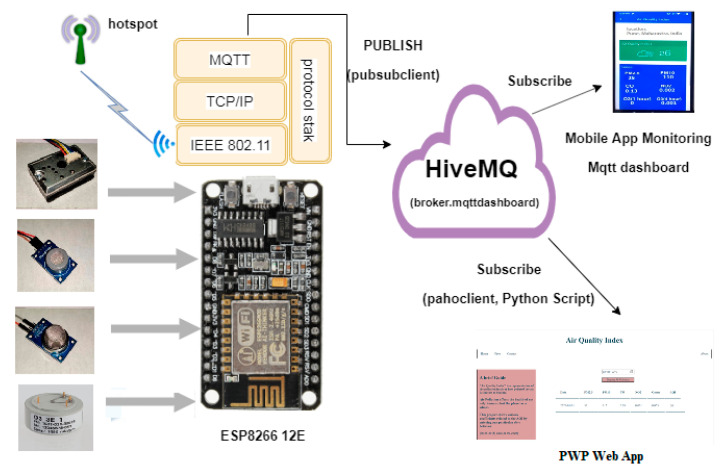
The PWP system architecture.

**Figure 4 sensors-20-05448-f004:**
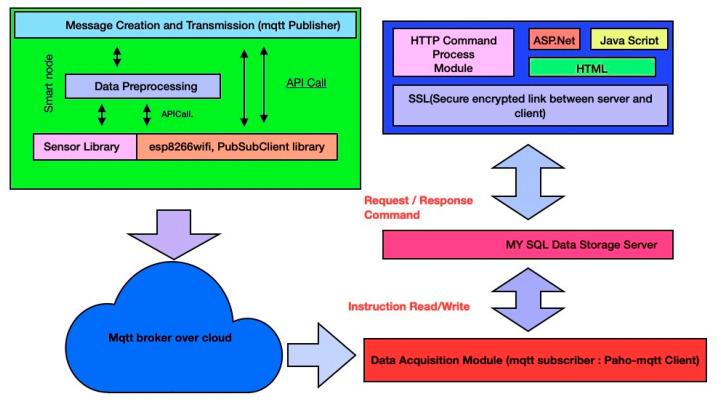
A detailed communication workflow of a PWP communication layer.

**Figure 5 sensors-20-05448-f005:**
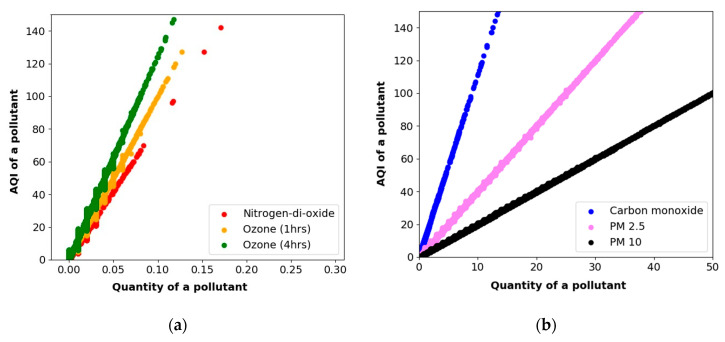
Linear relationship representations of a variety of pollutants: (**a**) nitrogen-dioxide (NO_2_), O_3_ (1 h), O_3_ (4 h) (**b**) PM10, and carbon monoxide (CO).

**Figure 6 sensors-20-05448-f006:**
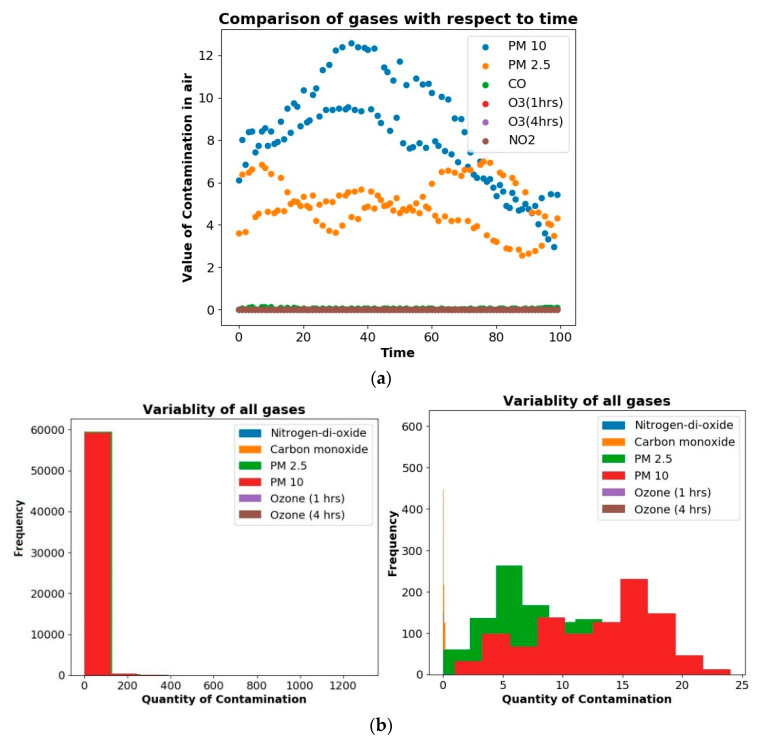
(**a**) A representation of the relationship between pollutants vs. time. (**b**) A variability analysis representation of the air pollutants PM2.5, PM10, CO, NO_2_, and O_3_.

**Figure 7 sensors-20-05448-f007:**
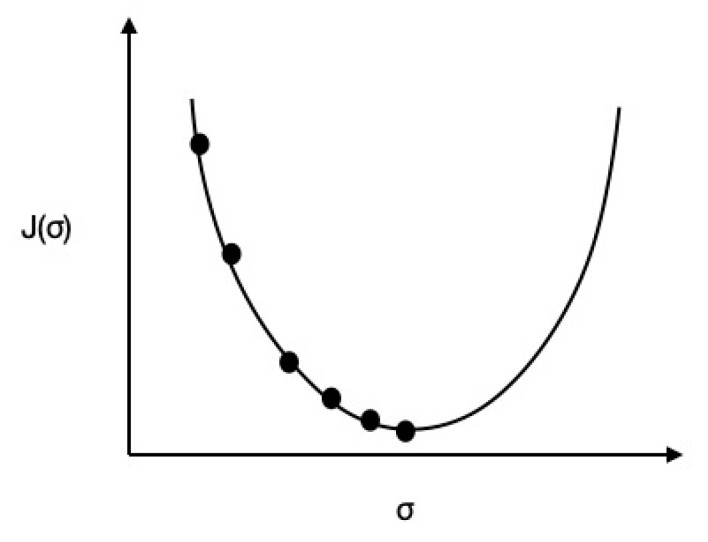
Descendant graph of j (σ ).

**Figure 8 sensors-20-05448-f008:**
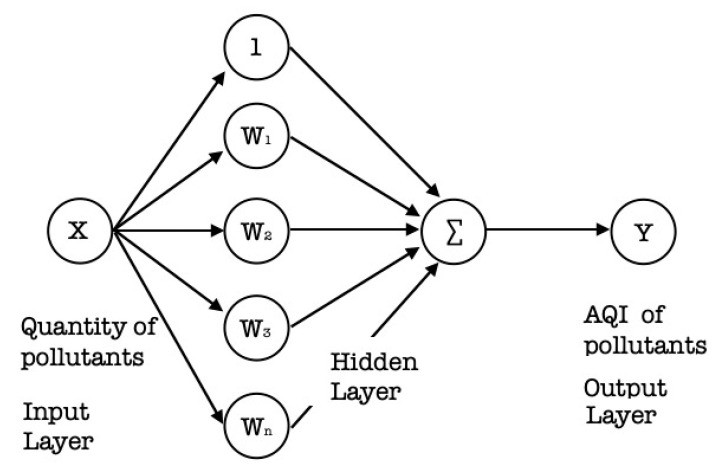
The customized artificial neural network (ANN) model of a PWP system.

**Figure 9 sensors-20-05448-f009:**
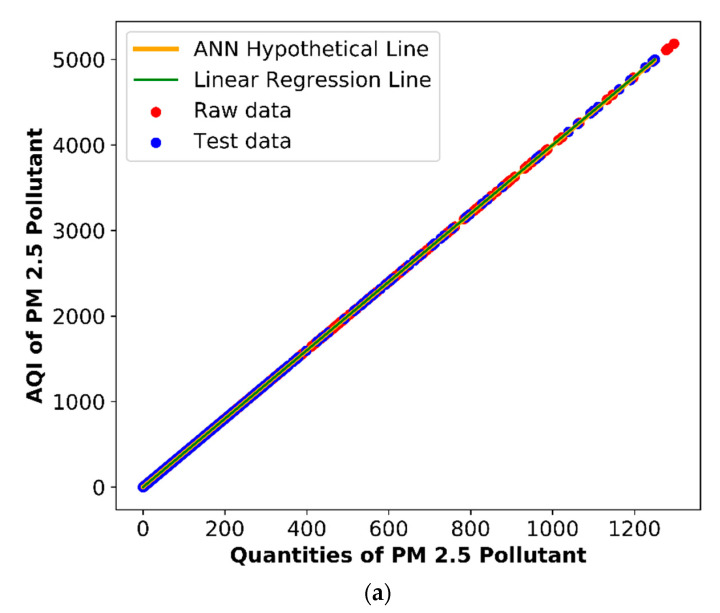
Linear regression and ANN-based comparative analysis of the air pollutants: (**a**) PM2.5, (**b**) PM10, (**c**) CO, (**d**) NO_2_, (**e**) O_3_ (1 h), and (**f**) O_3_ (4 h). AQI: air quality index.

**Figure 10 sensors-20-05448-f010:**
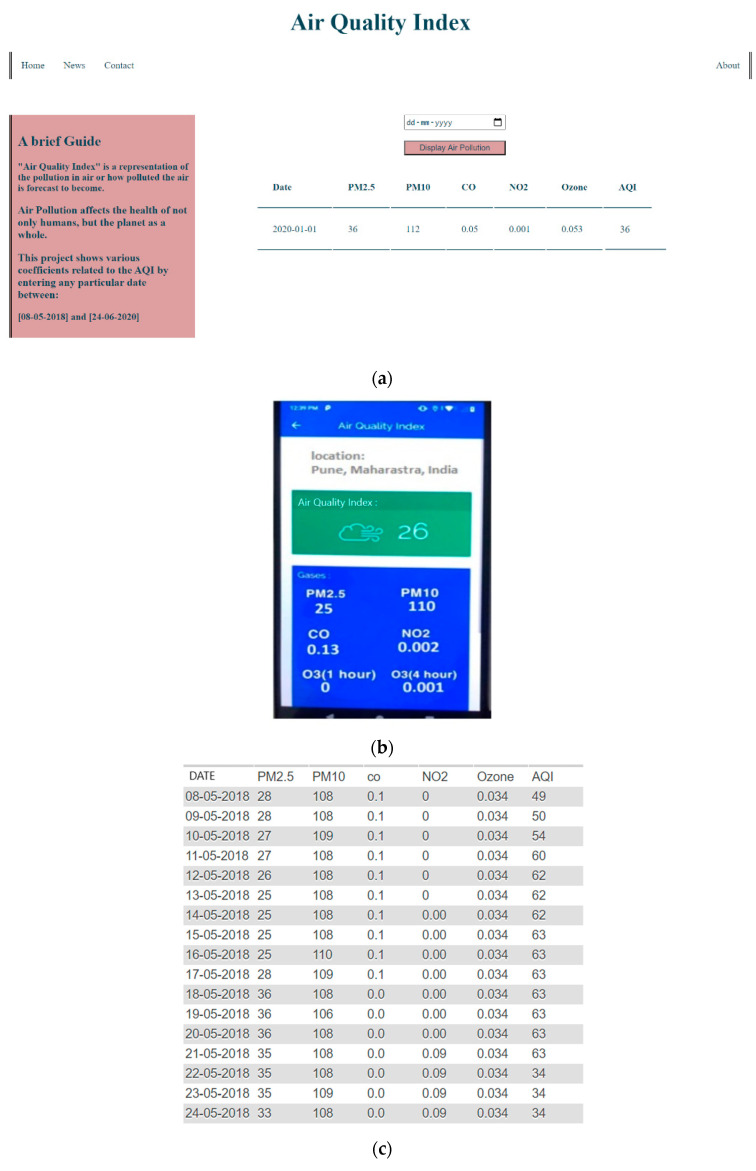
A representation of a PWP system. (**a**) Web interface, (**b**) mobile interface, and (**c**) date-wise pollution prediction report.

**Table 1 sensors-20-05448-t001:** A comparative analysis of various controllers [58].

Name	CPU Configurations	Price (Rs.)	Analog I/O Pins	Digital I/O Pins
Raspberry Pi 3	Quad Core1.2 GHz Broadcom BCM2837 64-bit Processor (1 GB, 42 g)	3200	0	14
Node MCU	Tensilica XtensaLX106 32-bit controller Wi-Fi SOC integrated (4 MB, 8 g)	400	1	17
BeagleBone Black	AM335x,1GHz ARM Cortex A8 Processor (512 MB, 40 g)	6000	6	14
Udoo (Quad)	Freescale MX6 Quad, 4 x Cortex™ A9 Core@ 1GHz processor With Atmel 32-bit cortex controller (1 GB, 150 g)	9000	14	62 + 14
Intel Galileo	Quark ™ X1000 32-bit 400 MHz (256 B, 370 g)	7000	6	14

**Table 2 sensors-20-05448-t002:** Technical specification details of the SDS021 sensing unit.

Features	Technical Specifications
Pollutants Measuring Capability	PM2.5 and PM10 levels
Sensitivity	0.001 mg/m^3^
Measurement range	0–100 mg/m^3^
Response time	N.A.
Zero drift	Auto (zero every hour)
Power backup	5 V
Related current	60 mA
Sleep Current	<4 mA
Temperature Range	−20 to +60 degrees
Threshold (minimum resolution of particles)	0.3 μm
Air Pressure	86 KPa to 110 KPa
Protocol	UART

**Table 3 sensors-20-05448-t003:** Technical specification details of a MQ07-CO (carbon monoxide) sensing unit.

Features	Technical Specifications
Pollutants Measuring Capability	CO levels
Sensitivity	−400 to −650 nA/ppm at 2 ppm
Measurement range	0–1000 ppm
Response time	<15 s
Zero drift	<0.1 ppm/year
Power backup	5 V
Related current	60 mA
Sleep Current	<4 mA
Temperature Range	−20 to +50 degrees
Relative humidity	<95%
Heating Consumption	350 mW
Protocol	UART

**Table 4 sensors-20-05448-t004:** Technical specification details of a NO2-B43F sensing unit.

Features	Technical Specifications
Pollutants Measuring Capability	NO_2_ levels
Sensitivity	−250 to −600 nA/ppm at 2 ppm
Response time	<25 s
Power backup	5 V
Measurement Range	0.001 to 0.02 ppm
Zero drift	0 to 0.02 ppm/year
Weight	<13 g
Temperature Range	3 to 20 degrees
Storage Period	6 months
Load resistor	33 to 100 Ω
Noise	±2 standard deviations (ppm equivalent)

**Table 5 sensors-20-05448-t005:** Technical specification details an Aeroqual Ozone (O_3_) sensing unit.

Features	Technical Specifications
Pollutants Measuring Capability	Ozone levels
Sensitivity	0.002 ppm
Power backup	3.3 V
Response Time	20 s
Measurement range	0.002 to 0.01 ppm
Temperature Range	−20 to +60 degrees
Zero drift	<0.02 ppm/day

**Table 6 sensors-20-05448-t006:** A representation of the classification of the air quality index (AQI) risk metrics [64].

AQI Values	Classification of Risk
0–50	Good
51–100	Moderate
101–150	Unhealthy for sensitive groups
151–200	Unhealthy
201–300	Very unhealthy
300 and above	Hazardous

**Table 7 sensors-20-05448-t007:** Details of the data collection for the conducted experiments.

Data Type	Number of Rows	Number of Columns	Status
Raw data	189,648	18	Raw data
Processed data	600,00	13	Feature selection (dimensionality reduction)
Missing data (1 value/row)	129,648	13	Removing Missing values
Training data	400,00	13	Data extraction for Training the model
Testing data	200,00	13	Data extraction for Testing the model

**Table 8 sensors-20-05448-t008:** Accuracy metrics of the linear regression methodology.

Pollutants	Weight Parameter(W)	Prediction Accuracy (%)
PM2.5	734.33	99.01
PM10	11.08	98.12
CO	943.78	97.79
NO_2_	1176.20	91
O_3_ (1 h)	4.00	96.21
O_3_ (4 h)	2.00	96.59

**Table 9 sensors-20-05448-t009:** Accuracy metrics of the artificial neural network (ANN) algorithm.

Pollutants	Prediction Accuracy (%)
PM2.5	90
PM10	84
CO	83
NO_2_	84.79
O_3_ (1 h)	79
O_3_ (4 h)	77

**Table 10 sensors-20-05448-t010:** A comparative analysis of the accuracy evaluation metrics for the linear, ridge, Lasso, Bayes, Huber, Lars, Lasso-Lars, stochastic gradient descent (SGD), ElasticNet, and ANN regression methodologies.

Methodologies	PM2.5 (%)	PM10 (%)	CO (%)	NO_2_ (%)	O_3_ (1 h (%))	O_3_ (1 h (%))
Linear Regression	99.01	98.12	97.79	91	96.21	96.59
Ridge Regression	98.11	97.68	97.45	89	94.34	93.21
Lasso Regression	97.23	97.11	97.33	87.78	93.98	93.12
Bayes Regression	97.32	96.59	97.08	86.78	92.79	93.02
Huber Regression	96.88	96.45	96.67	85.04	90.12	91.11
Lars Regression	96.67	96.55	96.01	84.49	89.98	90.34
Lasso-Lars Regression	95.56	95.54	94.99	80.76	88.77	90.11
SGD Regression	94.86	94.33	93.79	80.03	87.79	90.01
ElasticNet	94.78	94.13	93.54	79.76	87.16	89.78
ANN Regression	90	84	83	84.79	79	77

**Table 11 sensors-20-05448-t011:** A comparative analysis of the error metrics for the linear, ridge, Lasso, Bayes, Huber, Lars, Lasso-Lars, SGD, ElasticNet, and ANN regression methodologies.

Methodologies	Mean Absolute Error (%)	Mean Squared Error (%)
Linear Regression	1.12	3.22
Ridge Regression	1.23	3.23
Lasso Regression	1.40	4.20
Bayes Regression	1.77	4.23
Huber Regression	2.67	4.67
Lars Regression	3.33	5.23
Lasso-Lars Regression	4.06	6.63
SGD Regression	5.12	9.23
ElasticNet	6.02	10.20
ANN Regression	11.23	21.63

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
