# Peer review of "Pollution Weather Prediction System: Smart Outdoor Pollution Monitoring and Prediction for Healthy Breathing and Living"

_sensors, 2020, doi:10.3390/s20185448_

Round 1
Reviewer 1 Report
This paper reports linear regression and ANN methods for pollution weather prediction using data acquired in 90 days. My concern is the conclusion that “linear regression” is more suitable than ANN.
The conclusion is based on the ANN structure of 7 layers with numbers of neurons in five hidden layers as 256, 128, 64, 32 and 16. There are lots of questions related to the ANN structure: why 7 layers are needed? What the size of the training data set required for this specific ANN? These questions will lead to an assumption that this specific ANN might not work as good as linear regression, but it cannot support your claim since there might be a better ANN structure you have not discovered.
Other minor issues:
1. Remove source code start from line 427 to make the submission more like a technical report rather than a paper.
2. Redraw Fig.4. Fig.9 and Fig.10 (e.g. by merging subplots) to eliminate repeated patterns.
Author Response
Rebuttal- Reviewer 1
Dear Reviewer,
Thank you for giving us valuable suggestions to improve the presented research work. As per the given suggestions, we have incorporated all the changes and highlighted the changes in the blue color and figure, and table-caption changes have been represented using yellow color.
Q-1 My concern is the conclusion that “linear regression” is more suitable than ANN. The conclusion is based on the ANN structure of 7 layers with numbers of neurons in five hidden layers as 256, 128, 64, 32 and 16. There are lots of questions related to the ANN structure: why 7 layers are needed? What the size of the training data set required for this specific ANN? These questions will lead to an assumption that this specific ANN might not work as good as linear regression, but it cannot support your claim since there might be a better ANN structure you have not discovered.
Answer: Thank you for providing us a valuable suggestion to emphasize on core parts of the undertaken study. As per the given suggestions, we have incorporated the required changes in the ANN methodology, conclusion, and results section.
Q-2 Remove source code start from line 427 to make the submission more like a technical report rather than a paper.
Answer: Thank you for providing valuable suggestions related to representations. As per the give suggestions, we have removed a coding snapshot (Figure 8) for better visibility.
Q-3 Redraw Fig.4. Fig.9 and Fig.10 (e.g. by merging subplots) to eliminate repeated patterns.
Answer: Thank you for giving us valuable suggestions. As per the given suggestions, we have merged Fig. 4(a-e) in 2 images 5a and 5b for avoiding repetition of the patterns. Similarly, we have also merged Fig. 9 and 10 in Fig. 9 and represented comparisons between linear regression, other machine learning methodologies and the customized ANN algorithm.

Reviewer 2 Report
In general, the introduction to air pollution and its impact on human health as well as the introduction to the gap in the field of predicting air pollution concentrations that their paper is trying to fill feel scattershot and uninformative. It needs to be reworked so that the disparate ideas within the text are more cohesively tied together and more accurately explained, and so that the reader immediately understands what the author is trying to convey -- air pollution is bad because X, Y, Z; AQI has been used to communicate the health risk of air pollution concentrations but is imperfect; new advanced statistical methods can help; for example, we present the Pollution Weather System. I have provided a few specific comments to this effect, but altogether I suggest a full reconsideration and revision of content and presentation.
The English grammar needs a great deal of revision to improve the clarity of the authors' ideas and analyses and their relevance to the topic at hand.
What are the details of the 90-day exposure study? More context is needed for understanding the behavior of pollutants and sensors and the appropriateness of specific prediction methods employed in this paper. Was a human subjects review board involved in data collection?
The strengths, importance, or usefulness of AQI as a pollution risk metric should be more clearly explained – why did you chose this metric to predict?
A brief summary description of the device that includes all of the sensors together (listed in Table 1a and Section 2.1) is needed. Perhaps a few sentences before section 2.1.1.
Optical PM sensors – and perhaps the other gaseous sensors – often require considerable calibration as the optical properties of aerosols change over the course an hour, day, week, season, etc. How is this effect accounted for? If not, a paragraph pointing out this weakness in low-cost sensors is needed -- perhaps in the Discussion.
A flow diagram describing the data sample sizes from measurement, to training, to testing (etc.) along with data missingness would be very useful.
Could you explain why or how you decided on the ANN parameters you chose (e.g., relu, 7 layers, the hidden layer sizes, learning rate of 0.2, etc.)?
Line-specific commentary:
- Lines 37-42 (ending at “… mentioned above.”) may not be necessary; removal would make the document more succinct.
- Lines 44, 58: “WHO Reports” is not a specific enough resource, and the reference items “1” and “2” could be more rigorous and primary. Cite the specific reports from which the information came or, perhaps, one or more of the recent Lancet articles on IHME’s global burden of disease study (this is the other major reoccurring global GBD study): https://www.thelancet.com/gbd, especially https://www.thelancet.com/journals/lancet/article/PIIS0140-6736(18)32225-6/fulltext. Also see: https://www.stateofglobalair.org/report
- Line 49: “80% of urban breath air” is unclear. Do you mean “80% of urban populations breathe air”?
- Line 49: WHO “limits” should be more appropriately referred to as “guidelines.”
- Line 51 (And elsewhere): define all acronyms the first time they are used, e.g. “AQI”. Also, AQI is focused on in this paragraph as if it were a primary air pollutant rather than a mathematical construct created to provide information on the relative dangers of the concentrations of a variety of specific primary pollutants.
- Line 53: be more descriptive with “a fair amount” – does this mean a lot, a little, a moderate amount, etc.?
- Line 55: Rework “can reach the human system with ease” with more specificity.
- Line 60: I believe we know how PM2.5 is formed quite well under many circumstances. Do you mean that concentrations are difficult to predict?
- Line 61: The point of Annenberg et al. is that there are pollutants that affect both climate and health (“co-benefits”), and that BC and methane are two great examples of this. It is not that BC and methane are, specifically, the worst pollutants out there for health.
- Line 80: “far better” at what?
- Lines 99-155: This is a confusing section. Are you pointing to work that the Pollution Weather System improves upon, or are you demonstrating examples of other methods of air quality prediction, or both? It is unclear, and seems like it could be made into multiple paragraphs that connect an entire narrative, rather than jumping around between failures in the field vs. successes in the field upon which your work builds vs. approaches that are similar to yours. Moreover, it reads more like a Discussion Section than an Introduction. Perhaps some of this content is better suited for the Discussion?
- Line 163: What is variability research? It does not seem to be a known field of study. Do you mean that your work focuses mainly on measurement (or some other type of) variability?
- Line 173: “plagued with issues” should have at least a few high-level examples.
- Section 2.1: all sensors should have listed their lower and upper detection thresholds, measurement precision and resolution; name of measurement tech (e.g. optical particle counter, electrochemical cell, etc). Measurement accuracy as shown by third party tests where possible (e.g. http://www.aqmd.gov/aq-spec ) would be very useful, as well.
- Line 242: temperature is not a pollutant. It is an environmental factor.
- Figure 3: not all components of this figure seem to be defined or addressed in the paper’s text.
- Line 268: “AQI indexes” is redundant, as AQI stands for Air Quality Index.
- Lines 271 and 389: Is “1,89,648” a typo or the Indian convention for “189,648”? I believe the 1,89,648 convention is specific to India and may confuse non-Indian readers.
- Line 275: confusing. Are “training examples” further divided into training and testing “phases”?
- Line 276: Which specific dimensionality reduction technique?
- Section 2.2.3: a table identifying the specific model features used is needed.
- Line 283: Which scaling technique, and how are the data already organized in a way that makes the scaling technique unnecessary?
- Line 287, 312, 319: What does it variability in “µm” supposed to be a mass concentration like µg/m3? If so, was the output from sensors that report in ppm and ppb converted to the appropriate corresponding mass concentration? The “100” does not seem to correspond to a number on the graphics. Generally, do you mean categories are created of size 100 µg/m3 and 25 µg/m3 (or something along these lines)?
- Lines 289-305: The point of this visualization as well as the data that underly it. Are these measurements taken in the field [if so, which measurements] vs. predicted AQI? Are they just graphical representations of the mathematical AQI formula?
- Lines 370-373 seem like they should be Results, if these are the parameter values you arrived at using the gradient descent methods you describe.
- Line 436: Linear “regression”?
- Line 516: “Particulate” matter?
- Line 541-542 (and again on 579-581) is far too strong of a generalization and very likely untrue in many cases.
- Figure 11 suggests the system is predicting pollution concentrations and not AQI, which is antithetical to how the rest of the paper reads. Moreover, are these predictions retrospective or forecasts? Overall, the presentation of methods and results is quite confusing.
Author Response
Rebuttal- Reviewer 2
Dear Reviewer,
Thank you for giving us valuable suggestions to improve the presented research work. As per the given suggestions, we have incorporated all the changes and highlighted the changes in the red color and figure, and table-caption changes have been represented using yellow color.
Q-1 In general, the introduction to air pollution and its impact on human health as well as the introduction to the gap in the field of predicting air pollution concentrations that their paper is trying to fill feel scattershot and uninformative. It needs to be reworked so that the disparate ideas within the text are more cohesively tied together and more accurately explained, and so that the reader immediately understands what the author is trying to convey -- air pollution is bad because X, Y, Z; AQI has been used to communicate the health risk of air pollution concentrations but is imperfect; new advanced statistical methods can help; for example, we present the Pollution Weather System. I have provided a few specific comments to this effect, but altogether I suggest a full reconsideration and revision of content and presentation.
Answer: Thank you for giving valuable suggestions. Thank you for giving us valuable suggestions. As per the given suggestions, we have revised the introduction section by highlighting the suggested comments.
Q-2 The English grammar needs a great deal of revision to improve the clarity of the authors' ideas and analyses and their relevance to the topic at hand.
Answer: Thank you for giving valuable suggestions. As per the given suggestions, we have proof-read the whole manuscript and made the required changes throughout the manuscript including the introduction section. We have also highlighted the changes in the red color.
Q-3 What are the details of the 90-day exposure study? More context is needed for understanding the behavior of pollutants and sensors and the appropriateness of specific prediction methods employed in this paper. Was a human subjects review board involved in data collection?
Thank you for giving us valuable advice. As per the given suggestions, we have added the required details.
Q-4 The strengths, importance, or usefulness of AQI as a pollution risk metric should be more clearly explained – why did you chose this metric to predict?
Thank you for giving us valuable advice. As per the given suggestions, we have included an AQI risk metric table 6 and describe the same in section 2.2.3.
Q-5 A brief summary description of the device that includes all of the sensors together (listed in Table 1a and Section 2.1) is needed. Perhaps a few sentences before section 2.1.1.
Thank you for giving us valuable comments. As per the given comments, we have added the required configuration details as described in Table 1 to Table 5 and section 2.1, subsection 2.1.1 to 2.1.5. We have also added a Figure 3 which represents the system architecture of a PWP system.
Q-6 Optical PM sensors – and perhaps the other gaseous sensors – often require considerable calibration as the optical properties of aerosols change over the course an hour, day, week, season, etc. How is this effect accounted for? If not, a paragraph pointing out this weakness in low-cost sensors is needed -- perhaps in the Discussion
Thank you for giving us valuable comments. As per the given comments, we have added the required configuration details as described in Table 1 to Table 5 and section 2.1, subsection 2.1.1 to 2.1.5. We have also added a Figure 3 which represents the system architecture of a PWP system.
Q-7 A flow diagram describing the data sample sizes from measurement, to training, to testing (etc.) along with data missingness would be very useful.
Thank you for giving us valuable suggestions. As per the given suggestions, we have added the required details in the manuscript as shown in Table 7.
Q-8 Could you explain why or how you decided on the ANN parameters you chose (e.g., relu, 7 layers, the hidden layer sizes, learning rate of 0.2, etc.)?
Thank you for giving us valuable suggestions. As per the given suggestions, we have added the required details in the manuscript.
Line-specific commentary:
Q-9 Lines 37-42 (ending at “… mentioned above.”) may not be necessary; removal would make the document more succinct.
Thank you for giving us valuable suggestions. As per the given suggestions, we have removed the required text.
Q-10 Lines 44, 58: “WHO Reports” is not a specific enough resource, and the reference items “1” and “2” could be more rigorous and primary. Cite the specific reports from which the information came or, perhaps, one or more of the recent Lancet articles on IHME’s global burden of disease study (this is the other major reoccurring global GBD study): https://www.thelancet.com/gbd, especially https://www.thelancet.com/journals/lancet/article/PIIS0140-6736(18)32225-6/fulltext. Also see: https://www.stateofglobalair.org/report
Thank you for giving us valuable suggestions. As per the given suggestions, we have added the required references in the introduction and reference sections.
Q-11 Line 49: “80% of urban breath air” is unclear. Do you mean “80% of urban populations breathe air”?
Thank you for giving valuable suggestions. As per the given suggestions, we have the text to 80% of urban populations breathe air.
Q-12 Line 49: WHO “limits” should be more appropriately referred to as “guidelines.”
Thank you for giving valuable suggestions. As per the given suggestions, we have changed limits keyword proof-read the whole manuscript and made the required changes throughout the manuscript including the introduction section. We have also highlighted the changes in the red color.
Q-13 Line 51 (And elsewhere): define all acronyms the first time they are used, e.g. “AQI”. Also, AQI is focused on in this paragraph as if it were a primary air pollutant rather than a mathematical construct created to provide information on the relative dangers of the concentrations of a variety of specific primary pollutants.
Q-14 Line 53: be more descriptive with “a fair amount” – does this mean a lot, a little, a moderate amount, etc.?
Q-15 Line 55: Rework “can reach the human system with ease” with more specificity.
Thank you for giving us valuable suggestion. As per the given suggestions, we have made the required changes.
Q-16
Line 60: I believe we know how PM2.5 is formed quite well under many circumstances. Do you mean that concentrations are difficult to predict?
Thank you for giving us valuable suggestion. As per the given suggestions, we have made the required changes.
Q-17 Line 61: The point of Annenberg et al. is that there are pollutants that affect both climate and health (“co-benefits”), and that BC and methane are two great examples of this. It is not that BC and methane are, specifically, the worst pollutants out there for health.
Thank you for giving us valuable suggestions. As per the given suggestions, we have made the required changes.
Q-18 Line 80: “far better” at what?
Thank you for giving us valuable suggestions. As per the given suggestions, we have made the required changes.
Q-19
Lines 99-155: This is a confusing section. Are you pointing to work that the Pollution Weather System improves upon, or are you demonstrating examples of other methods of air quality prediction, or both? It is unclear, and seems like it could be made into multiple paragraphs that connect an entire narrative, rather than jumping around between failures in the field vs. successes in the field upon which your work builds vs. approaches that are similar to yours. Moreover, it reads more like a Discussion Section than an Introduction. Perhaps some of this content is better suited for the Discussion?
Thank you for giving us valuable suggestions. As per the given suggestions, we have made the required changes in the introduction and related work section.
Q-20 Line 163: What is variability research? It does not seem to be a known field of study. Do you mean that your work focuses mainly on measurement (or some other type of) variability?
Thank you for giving us valuable suggestions. We interpret variability as a measurement of air pollutant AQI values and its variations as described in section 2.2.3. However, to avoid confusions, we have made the required changes in section 2.
Q-21 Line 173: “plagued with issues” should have at least a few high-level examples.
Thank you for giving us valuable comments. As per the given comments, we have made the required changes.
Q-22 Section 2.1: all sensors should have listed their lower and upper detection thresholds, measurement precision and resolution; name of measurement tech (e.g. optical particle counter, electrochemical cell, etc). Measurement accuracy as shown by third party tests where possible (e.g. http://www.aqmd.gov/aq-spec) would be very useful, as well.
Thank you for giving us valuable comments. As per the given comments, we have added the required configuration details as described in Table 1 to Table 5 and section 2.1, subsection 2.1.1 to 2.1.5.
Q-23 Line 242: temperature is not a pollutant. It is an environmental factor
Thank you for giving us valuable comments. As per the given comments, we have removed the word temperature which was written by mistake.
Q-24 Figure 3: not all components of this figure seem to be defined or addressed in the paper’s text
Thank you for giving us valuable comments. As per the given comments, we have added the details of all the components and its interface. The Figure 3 has been changed to Figure 4.
Q-25 Line 268: “AQI indexes” is redundant, as AQI stands for Air Quality Index.
Thank you for giving us valuable comments. As per the given comments, we have removed the word ‘indexes’ and changed it to “values”.
Q-26 Lines 271 and 389: Is “1,89,648” a typo or the Indian convention for “189,648”? I believe the 1,89,648 convention is specific to India and may confuse non-Indian readers. 189,648
Thank you for giving us valuable comments. As per the given comments, we have made the required changes.
Q-27 Line 275: confusing. Are “training examples” further divided into training and testing “phases”?
Thank you for giving us valuable suggestions. As per the given comments, we have made the required changes.
Q-28 Line 276: Which specific dimensionality reduction technique?
Thank you for giving us valuable suggestions. As per the given comments, we have included the feature selection dimensionality reduction technique and also given more details in Table 7.
Q-29 Section 2.2.3: a table identifying the specific model features used is needed
Thank you for giving us valuable suggestions. As per the given comments, we have included Table 7 to highlight model features and techniques.
Q-30 Line 283: Which scaling technique, and how are the data already organized in a way that makes the scaling technique unnecessary?
Thank you for giving us valuable suggestions. We have made the required changes. In the conducted experiments standard-scaler scaling technique has been used.
Q-31 Line 287, 312, 319: What does it variability in “µm” supposed to be a mass concentration like µg/m3? If so, was the output from sensors that report in ppm and ppb converted to the appropriate corresponding mass concentration? The “100” does not seem to correspond to a number on the graphics. Generally, do you mean categories are created of size 100 µg/m3 and 25 µg/m3 (or something along these lines)?
Thank you for giving us valuable comments. As per the given comments, we have added the required conversion formula as described in eq(2) and eq(3) and section 2.2.3.
Q-32 Lines 289-305: The point of this visualization as well as the data that underly it. Are these measurements taken in the field [if so, which measurements] vs. predicted AQI? Are they just graphical representations of the mathematical AQI formula?
Thank you for giving us valuable comments. As per the given comments, we have added the required formula of AQI as described in eq(1) and section 2.2.3.
Q-33 Lines 370-373 seem like they should be Results, if these are the parameter values you arrived at using the gradient descent methods you describe.
Thank you for giving us valuable suggestions. Yes, the parameter values have been calculated using gradient descent methods as described in 2.2.4a.
Q-34 Line 436: Linear “regression”?
Thank you for giving us typo corrections. We have made the required changes.
Q-35 Line 516: “Particulate” matter?
Thank you for giving us typo corrections. We have made the required changes.
Q-36 Line 541-542 (and again on 579-581) is far too strong of a generalization and very likely untrue in many cases.
Thank you for giving us valuable suggestions. As per the given comments, we have made the required changes. We have done re-experimentation and recorded better accuracy with the customized ANN regression methodology.
Q-37 Figure 11 suggests the system is predicting pollution concentrations and not AQI, which is antithetical to how the rest of the paper reads. Moreover, are these predictions retrospective or forecasts? Overall, the presentation of methods and results is quite confusing.
Thank you for giving us valuable suggestions. As per the given comments, we have updated the required images as shown in Figure 10.

Round 2
Reviewer 1 Report
no further comments
Author Response
Dear Reviewer,
Thank you for spending time for reviewing and accepting our manuscript.
Reviewer 2 Report
The AQI needs a mathematical description (perhaps an example of just the calculation of AQI for PM2.5).
How far into the future, if at all, are you predicting AQI values? Predicting AQI values from actual measured concentrations is literally what the AQI is – a mathematical conversion of concentration values. If your major contribution is that you can use low cost sensors to accurately produce the same AQI values as very expensive monitors (like regulatory monitors), then that needs to be made very clear. If you are predicting future values and creating a forecast, then that also needs to be clear too (or instead). Or, if neither, and you are simply stating an interesting finding that one can model AQI in this unique set of ways (which I think is truly interesting), then state as much early on in the introduction and in the Discussion section.
Has anyone else presented on this interesting finding – that linear regression of low-cost sensor data can be used to “predict” AQI with quite minimal error?
Section 2.2.3 needs much more information on the collection of data over 90 days. Where are data for PWP collected – which city and under which seasons? Where is/are monitoring station(s) located? Is this part of a larger data collection effort that is reported on elsewhere? This is critical for understanding the context for interpretation and validity of the model you present.
What is considered “truth” for these predictions? Are you comparing data against government reported AQI values at some specific location?
The introduction is still long (it should be reduced considerably) and jumps around from topic to topic too much. Each topic should be its own paragraph or cluster of sentences. E.g., the health effects of different air pollutants. I suggest five paragraphs or paragraph clusters:
- Why air pollution is bad and, relative to other risk factors, how does air pollution exposure in general and for each specific pollutant that you will study compare.
- What the AQI is (a metric for normalizing and communicating the risks associated with concentrations of different pollutants – for each pollutant and in aggregate).
- What your study does (predict AQI) and why you chose the AQI as your metric of interest.
- What it adds to the relevant literature – major contributions to the literature e.g. starting at Line 200
- Where it succeeds or adds where other studies have failed or omitted. Perhaps this entire bit, which is roughly Lines 95 – 177, belongs as part of the Discussion.
Discussion of cross-sensitivities of each sensor used would add context to potential external biasing factors in/of your models.
- Line 38: “… is less of development and more of a risk…” seems pedantic and maybe overstated. I’d recommending saying something like “…development provides benefits but it also poses risks…”
- Line 42: the GBD is most typically referred to as “the Global Burden of Disease study” – no pluralization on “disease,” and no inclusion of “Injuries, and Risk Factors”
- Line 42-46: GBD requires citation.
- Line 45: “daily” should be “DALYs”
- Line 49: “worse” should be “worst”
- Line 54: “hair”?
- Line 59: Suggested edit: “…can be microscopic and reach the…”
- Line 60: Vary in what kind of range? Presumably this means “can vary in size”?
- Line 69: Kyrkilis study needs more background – what 18 cities? Where were the cities – in India? 5 out of 18 cities had higher pollution than what? Than the other 13 cities? And those other 13 studies also had bad air quality? This is a confusing addition.
- Line 76-78. Both the point and the meaning of this addition are unclear. Specifically “…,which then results in knowing the formation process of the pollutant depending on the methodology.” Perhaps this citation belongs in a different paragraph?
- Lines 80-82. Do not belong in this paragraph, and may be removed completely.
- Line 102. “Despite…” can start a new paragraph.
- Line 106: For which pollutants is AQI calculated? Are any missing from your sensor array?
- Line 112 Kim et al seems to not fit in this paragraph. If you really want to discuss studies of health impacts of air pollutants, it should be its own paragraph or later in the Discussion.
- Line 145 – bracketed citations for recent high-profile mobile monitoring should be added
- Generally: the use of “the proposed approach” and “the presented approach” in this writeup is confusing, because it typically refers to the paper that you are writing (not a paper that you are citing)
- Lines 185-186: prediction of what? AQI as it relates to those pollutants? Specific concentrations of those pollutants?
- Line 204: how far into the future, if at all, are you predicting AQI values?
- Please check Tables 1-5 for correctness. For example, I found the following inconsistencies and wonder if there may be other errors in these tables:
- Line 247 states an operating range of 0.5 ppm, but Table 5 indicates a max of 1000 ppm.
- Line 265 states an operating range of 0.5 ppm, but Table 5 indicates a max of 10 ppb.
- Line 304: “Pollutants present are mainly…” should be “Pollutants measured are…”. There are many more pollutants “present” in the air, and a list of “main” pollutants would probably differ based on which scientists you asked.
- Line 353 / Equation 1: the first term cancels out to 1 as currently written. Pmax is not included in the equation as currently written.
- Equations 1, 2, and 3 and Table 6 should have citations.
- Line 368: Kelvin does not use degrees ( ° )
- Figure 5. Why does AQI go to 5000 in part b? Also in part b, CO does not show up very well. For parts a and b, why do different pollutants extend to different values along the Y (and x) axes? As an in-silica representation (or is this not the case?) I would expect all would go at least as far as the end of the frame or plot outline.
- Line 427: “o-7 µm” should be “0-7 µm” (zero is the letter air)
- Figure 6: The top 2 plots are the same, are they not? Also, these values are low for PM10 and PM2.5, and may very well be below the actual/practical detection limit of the low-cost PM sensors used. Also, are the data here from your low cost sensors or somewhere else? Where were they taken and what context do they represent (rush hour in some city downtown? etc.)
- Section 2.2.4: again, are these future values being predicted or are you comparing low-cost sensor-based AQI estimates to high-cost monitor based AQI calculations?
- Line 503: Is this supposed to include the unit that each pollutant uses, too?
- Lines 554-573: The analysis of the multi-pollutant model is interesting and seems reasonable if analyzing pollutant-specific models (vs. the model that predicts total AQI from all pollutants in aggregate). Perhaps it should be reconsidered and re-written to account for the fact that more than just concentration is at play – it is (At least) both the concentration of each pollutant as well as the frequency of variation and overall background presence of that pollutant.
- Line 557: Does “x-intercept” mean “slope” or “model coefficient”, or is it actually the point at which y (AQI) = 0 for each pollutant?
- Line 637 “podllutant”
- Line 650: should “86%” be “84%” – based on Table 9?
- Line 665: The fact that you explored all of these methods should be included in the paper’s Methods section. This is the first mention of them outside of the abstract.
- Section 4: This is probably better suited as supplemental text rather than a section after the Results and Discussion.
- Table 12 belongs in the Introduction
- Lines 739 & 741: By “some increases” do you mean “slight increases” or “small relative increases”?
- Section 5 should also include the intriguing finding that linear regression can be used to accurately “predict” AQI. Again, what you mean by “predict” should be very clear here and in the rest of the paper.
Author Response
Rebuttal- Reviewer 2
Dear Reviewer,
Thank you for giving us valuable suggestions to improve the presented research work. As per the given suggestions, we have incorporated all the changes and highlighted the changes in the blue color and figure, and table-caption changes have been represented using yellow color.
Q-1 How far into the future, if at all, are you predicting AQI values? Predicting AQI values from actual measured concentrations is literally what the AQI is – a mathematical conversion of concentration values. If your major contribution is that you can use low cost sensors to accurately produce the same AQI values as very expensive monitors (like regulatory monitors), then that needs to be made very clear. If you are predicting future values and creating a forecast, then that also needs to be clear too (or instead). Or, if neither, and you are simply stating an interesting finding that one can model AQI in this unique set of ways (which I think is truly interesting), then state as much early on in the introduction and in the Discussion section.
Thank you for giving us valuable suggestions. In the presented research work, we have used machine learning and deep neural network-based approaches to predict AQI values of various air pollutants such as PM 2.5, PM 10, CO, NO2, and O3. The presented approach can function lifelong and can be retrained according to the updated dataset.
Q-2 Has anyone else presented on this interesting finding – that linear regression of low-cost sensor data can be used to “predict” AQI with quite minimal error?
Thank you for giving us valuable inputs. According to my knowledge and based on the conducted literature work, we can assure that such an approach has not been used for the presented experiments. This was the primary motivation for conducting the presented research work.
Q-3 Section 2.2.3 needs much more information on the collection of data over 90 days. Where are data for PWP collected – which city and under which seasons? Where is/are monitoring station(s) located? Is this part of a larger data collection effort that is reported on elsewhere? This is critical for understanding the context for interpretation and validity of the model you present.
Thank you for giving us valuable inputs. We have made the required changes. In the presented research work, we have presented a PWP system that is configured with pollution sensing units such as SDS021, MQ07-CO, NO2-B43F, and Aeroqual Ozone(O3). These sensing units were utilized to collect and measure various pollutant levels such as PM 2.5, PM10, CO, NO2, and O3 for 90 days at Symbiosis International University, Pune, Maharashtra, India. The data collection was carried out between the duration of December 2019 to February 2020 during Winter.
Q-4 What is considered “truth” for these predictions? Are you comparing data against government reported AQI values at some specific location?
Thank you for giving us valuable inputs. In the presented research, we have formulated customized approach for calculating AQI values as described in section 2.2.3.
Q-5 The introduction is still long (it should be reduced considerably) and jumps around from topic to topic too much. Each topic should be its own paragraph or cluster of sentences. E.g., the health effects of different air pollutants. I suggest five paragraphs or paragraph clusters:
- Why air pollution is bad and, relative to other risk factors, how does air pollution exposure in general and for each specific pollutant that you will study compare.
- What the AQI is (a metric for normalizing and communicating the risks associated with concentrations of different pollutants – for each pollutant and in aggregate).
- What your study does (predict AQI) and why you chose the AQI as your metric of interest.
- What it adds to the relevant literature – major contributions to the literature e.g. starting at Line 200
- Where it succeeds or adds where other studies have failed or omitted. Perhaps this entire bit, which is roughly Lines 95 – 177, belongs as part of the Discussion.
Discussion of cross-sensitivities of each sensor used would add context to potential external biasing factors in/of your models.
Thank you for giving us suggestions. As per the given suggestions, we have divided introduction section in 5 separate paragraphs and made the required changes.
Q-6 Line 38: “… is less of development and more of a risk…” seems pedantic and maybe overstated. I’d recommending saying something like “…development provides benefits but it also poses risks…”
Thank you for giving us suggestions. We have made the required changes.
Q-7 Line 42: the GBD is most typically referred to as “the Global Burden of Disease study” – no pluralization on “disease,” and no inclusion of “Injuries, and Risk Factors”
Thank you for giving us suggestions. We have made the required changes.
Q-8 Line 42-46: GBD requires citation.
Thank you for giving us suggestions. We have added the required citation.
Q-9 Line 45: “daily” should be “DALYs”
Thank you for giving us formatting related suggestions. We have made the required changes.
Q-10 Line 49: “worse” should be “worst”
Thank you for giving us formatting related suggestions. We have made the required changes.
Q-11 Line 54: “hair”?
Thank you for giving us formatting suggestions. We have made the required changes.
Q-12 Line 59: Suggested edit: “…can be microscopic and reach the…”
Thank you for giving us formatting suggestions. We have made the required changes.
Q-13 Line 60: Vary in what kind of range? Presumably this means “can vary in size”?
Thank you for giving us formatting suggestions. We have made the required changes.
Q- 14 Line 69: Kyrkilis study needs more background – what 18 cities? Where were the cities – in India? 5 out of 18 cities had higher pollution than what? Than the other 13 cities? And those other 13 studies also had bad air quality? This is a confusing addition.
Thank you for giving us valuable suggestions. We have made the required changes.
Q-15 Line 76-78. Both the point and the meaning of this addition are unclear. Specifically, “…, which then results in knowing the formation process of the pollutant depending on the methodology.” Perhaps this citation belongs in a different paragraph?
Thank you for giving us valuable suggestions. We have made the required changes.
Q-16 Lines 80-82. Do not belong in this paragraph, and may be removed completely.
Thank you for giving us valuable suggestions. We have made the required changes.
Q-17 Line 102. “Despite…” can start a new paragraph.
Thank you for giving us valuable suggestions. We have made the required changes.
Q-18 Line 106: For which pollutants is AQI calculated? Are any missing from your sensor array?
Thank you for giving us valuable suggestions. We have calculated AQI values for a variety of pollutants such as PM 2.5, PM 10, CO, NO2, and O3. We have verified that we are not missing any of the presented pollutants.
Q-19 Line 112 Kim et al seems to not fit in this paragraph. If you really want to discuss studies of health impacts of air pollutants, it should be its own paragraph or later in the Discussion.
Thank you for giving us valuable suggestions. We have made the required changes and shifted the citation in an appropriate paragraph.
Q-20 Line 145 – bracketed citations for recent high-profile mobile monitoring should be added
Thank you for giving us valuable suggestions. We have added the required citations.
Q-21 Generally: the use of “the proposed approach” and “the presented approach” in this write-up is confusing, because it typically refers to the paper that you are writing (not a paper that you are citing)
Thank you for giving us valuable suggestions. We have made the required changes. We have made the use of words such as analysed, discussed, researched, studied, examined, conducted study etc. We have highlighted it with yellow color throughout the paper.
Q- 22 Lines 185-186: prediction of what? AQI as it relates to those pollutants? Specific concentrations of those pollutants?
Thank you for giving us valuable suggestions. We have made the required changes.
Q- 23 Line 204: how far into the future, if at all, are you predicting AQI values?
Thank you for giving us valuable suggestions. Yes, we are predicting AQI values.
Q- 24 Please check Tables 1-5 for correctness. For example, I found the following inconsistencies and wonder if there may be other errors in these tables:
Thank you for giving us valuable suggestions. We have made the required changes and highlighted table and figure captions with yellow color.
Q-25 Line 247 states an operating range of 0.5 ppm, but Table 5 indicates a max of 1000 ppm.
Thank you for giving us valuable suggestions. We have made the required changes.
Q-26 Line 265 states an operating range of 0.5 ppm, but Table 5 indicates a max of 10 ppb.
Thank you for giving us valuable suggestions. We have made the required changes.
Q-27 Line 304: “Pollutants present are mainly…” should be “Pollutants measured are…”. There are many more pollutants “present” in the air, and a list of “main” pollutants would probably differ based on which scientists you asked.
Thank you for giving us valuable suggestions. We have made the required changes.
Q-28 Line 353 / Equation 1: the first term cancels out to 1 as currently written. Pmax is not included in the equation as currently written.
Thank you for giving us valuable suggestions. We have made the required changes.
Q-29 Equations 1, 2, and 3 and Table 6 should have citations.
Thank you for giving us valuable suggestions. We have made the required citations in equations 1 to 3), and Table 6.
Q-30 Line 368: Kelvin does not use degrees ( ° )
Thank you for giving us valuable suggestions. We have made the required changes.
Q-31 Figure 5. Why does AQI go to 5000 in part b? Also in part b, CO does not show up very well. For parts a and b, why do different pollutants extend to different values along the Y (and x) axes? As an in-silica representation (or is this not the case?) I would expect all would go at least as far as the end of the frame or plot outline.
Thank you for giving us valuable suggestions. We have made the required changes in Figure 5.
Q-32 Line 427: “o-7 µm” should be “0-7 µm” (zero is the letter air)
Thank you for giving us valuable suggestions. We have made the required changes.
Q-33 Figure 6: The top 2 plots are the same, are they not? Also, these values are low for PM10 and PM2.5, and may very well be below the actual/practical detection limit of the low-cost PM sensors used. Also, are the data here from your low cost sensors or somewhere else? Where were they taken and what context do they represent (rush hour in some city downtown? etc.)
Thank you for giving us valuable suggestions. We have made the required changes in Figure 6. In the presented research work, we have presented a PWP system that is configured with pollution sensing units such as SDS021, MQ07-CO, NO2-B43F, and Aeroqual Ozone(O3). These sensing units were utilized to collect and measure various pollutant levels such as PM 2.5, PM10, CO, NO2, and O3 for 90 days at Symbiosis International University, Pune, Maharashtra, India. The data collection was carried out between the duration of December 2019 to February 2020 during Winter.
Q-34 Section 2.2.4: again, are these future values being predicted or are you comparing low-cost sensor-based AQI estimates to high-cost monitor based AQI calculations?
Thank you for giving us valuable suggestions. Here, the future values are being predicted.
Q-35 Line 503: Is this supposed to include the unit that each pollutant uses, too?
Thank you for giving us valuable suggestions. We have made the required changes.
Q-36 Lines 554-573: The analysis of the multi-pollutant model is interesting and seems reasonable if analysing pollutant-specific models (vs. the model that predicts total AQI from all pollutants in aggregate). Perhaps it should be reconsidered and re-written to account for the fact that more than just concentration is at play – it is (At least) both the concentration of each pollutant as well as the frequency of variation and overall background presence of that pollutant.
Thank you for giving us valuable suggestions. Furthermore, in the conducted experiments, we have only considered a relation between quantities of pollutants and AQI values of pollutants at an outdoor site (Symbiosis International University, Pune, Maharashtra, India) for 90 days. We have not considered variations such as places with high altitude, geographical variations, and Spatio-temporal variations. These are the future directions for fellow researchers.
Q-37 Line 557: Does “x-intercept” mean “slope” or “model coefficient”, or is it actually the point at which y (AQI) = 0 for each pollutant?
Thank you for asking important questions. It is our model coefficient.
Q-38 Line 637 “pollutant”
Thank you for giving us valuable suggestions. We have made the required changes.
Q-39 Line 650: should “86%” be “84%” – based on Table 9?
Thank you for giving us valuable suggestions. We have made the required changes.
Q-40 Line 665: The fact that you explored all of these methods should be included in the paper’s Methods section. This is the first mention of them outside of the abstract.
Thank you for giving us valuable suggestions. We have made the required changes.
Q-41 Section 4: This is probably better suited as supplemental text rather than a section after the Results and Discussion.
Thank you for giving us valuable suggestions. We have made the required changes.
Q-42 Table 12 belongs in the Introduction
Thank you for giving us valuable suggestions. We have shifted Table 12 to introduction section and captioned as Table 1.
Q-43 Lines 739 & 741: By “some increases” do you mean “slight increases” or “small relative increases”?
Thank you for giving us valuable suggestions. We have made the required changes.
Q-44 Section 5 should also include the intriguing finding that linear regression can be used to accurately “predict” AQI. Again, what you mean by “predict” should be very clear here and in the rest of the paper.
Thank you for giving us valuable suggestions. We have made the required changes. In the presented research work, we have used machine learning and deep neural network-based approaches to predict AQI values of various air pollutants such as PM 2.5, PM 10, CO, NO2, and O3.
